# Mutant p53, the Mevalonate Pathway and the Tumor Microenvironment Regulate Tumor Response to Statin Therapy

**DOI:** 10.3390/cancers14143500

**Published:** 2022-07-19

**Authors:** Madison Pereira, Kathy Matuszewska, Alice Glogova, Jim Petrik

**Affiliations:** Department of Biomedical Sciences, University of Guelph, Guelph, ON N1G 2W1, Canada; mperei02@uoguelph.ca (M.P.); kmatusze@uoguelph.ca (K.M.); aglogova@uoguelph.ca (A.G.)

**Keywords:** metabolism, p53, mevalonate pathway, therapy resistance, statins

## Abstract

**Simple Summary:**

For tumors to initiate and sustain rapid growth, they need to alter their metabolic reprograming to support this growth. While many oncogenes upregulate metabolic pathways in cancer, here, we focus on how mutant p53 interacts with the mevalonate pathway (MVA) and functions as a continual on switch, causing tumor cells to grow uncontrollably. The relationship of mutant p53 and the MVA pathway in promoting tumorigenesis and the pleiotropic effects within the tumor microenvironment as a result of MVA pathway upregulation are discussed in this review. Many individuals are prescribed a statin drug to decrease their circulating cholesterol levels by blocking the MVA pathway. Several studies have suggested the potential of repurposing statins to treat various cancers, since tumor cells upregulate and rely substantially on the MVA pathway. We evaluate previous studies on statins and cancer incidence and patient mortality, as well as future research to optimize the use of statins as a cancer therapy.

**Abstract:**

Tumor cells have the ability to co-opt multiple metabolic pathways, enhance glucose uptake and utilize aerobic glycolysis to promote tumorigenesis, which are characteristics constituting an emerging hallmark of cancer. Mutated tumor suppressor and proto-oncogenes are frequently responsible for enhanced metabolic pathway signaling. The link between mutant p53 and the mevalonate (MVA) pathway has been implicated in the advancement of various malignancies, with tumor cells relying heavily on increased MVA signaling to fuel their rapid growth, metastatic spread and development of therapy resistance. Statin drugs inhibit HMG-CoA reductase, the pathway’s rate-limiting enzyme, and as such, have long been studied as a potential anti-cancer therapy. However, whether statins provide additional anti-cancer properties is worthy of debate. Here, we examine retrospective, prospective and pre-clinical studies involving the use of statins in various cancer types, as well as potential issues with statins’ lack of efficacy observed in clinical trials and future considerations for upcoming clinical trials.

## 1. Cellular Metabolism of Normal and Cancerous Cells

### 1.1. Cellular Metabolism in Normal Non-Cancerous Cells

Normal, non-cancerous cells rely primarily on mitochondrial oxidative phosphorylation to maintain metabolic homeostasis. Under oxygenated, aerobic conditions, cells metabolize glucose through oxidative phosphorylation [1]. For this to occur, glycolysis facilitates the breakdown of glucose to pyruvate while releasing ATP and NADH [2]. In the mitochondria, pyruvate is converted into acetyl-CoA through the citric acid cycle, which yields ATP, NADH and FADH [2]. The reduced coenzymes (NADH and FADH) are what fuels the electron transport chain to undergo oxidative phosphorylation and generate ATP for the cell (Figure 1A) [3].

During anaerobic conditions, there is increased demand for rapid ATP production in order to fuel cells with ample energy due to a lack of oxygen [4]. Although this process, known as substrate-level phosphorylation, does not yield as much ATP compared to oxidative phosphorylation, it allows cells to sustain energy requirements until oxygen levels are replenished [4]. Under these conditions, the mitochondria are no longer capable of oxidizing the pyruvate molecules created through glycolysis [5]. Instead, pyruvate is converted to lactate to prevent its accumulation and promote the continuation of glycolysis [5]. NAD+ is subsequently regenerated in the cytosol and is utilized in glycolysis to sustain ATP production by cells (Figure 1B) [5]. When homeostatic oxygen levels restore, cells resume aerobic glycolysis coupled with oxidative phosphorylation.

### 1.2. The Role of Metabolism in Tumorigenesis 

The reprograming of energy metabolism by tumor cells is an established hallmark of cancer [6]. Alterations to cellular energy metabolism allow for support of chronic proliferation, which is facilitated primarily through increased consumption of glucose and glutamine [7]. The Warburg effect characterizes the phenomenon where tumor cells increase the rate of glucose uptake, with a preference for fermentation and production of lactate, regardless of the presence or absence of oxygen [8]. Preference for aerobic glycolysis, even when oxygen is present, is an adaptation by tumor cells exposed to intermittent hypoxic conditions as a result of vascular dysfunction and reduced perfusion [9,10]. Aerobic glycolysis is promoted due to activation of oncogenes, loss of tumor suppressors, upregulation of PI3K/Akt/mTOR and Ras/Raf/MEK/ERK pathways and downregulation of the LKB1-AMPK pathway [11]. As such, the accumulation of lactate from increased aerobic glycolysis facilitates acidosis within the tumor microenvironment (Figure 2) [12]. An acidic environment with associated low pH levels in the extracellular space fosters increased angiogenesis, invasiveness and metastasis of tumor cells and worse clinical outcomes for cancer patients [13]. Tumor cells are known to use an adaptive citric acid cycle in which most of the acetyl-CoA is produced from glucose, and many intermediate metabolites are created through glutaminolysis, where glutamine uptake results in the production of lactate (Figure 2) [14,15]. Mitochondrial adaptations also occur in order to be functional in a low-oxygen and acidic environment. Hypoxia-inducible factor 1-alpha (HIF-1α) has been shown to activate pyruvate dehydrogenase kinase 1 and promote glycolysis, as oxidative phosphorylation is slowed by hypoxic conditions [16]. Pyruvate dehydrogenase kinase 1 activity inhibits pyruvate dehydrogenase, contributing to a shortage of pyruvate availability for the tricarboxylic acid cycle [17]. In turn, this lack of pyruvate decreases mitochondrial oxygen consumption and increases intracellular oxygen availability, to ultimately promote tumor cell survival [18,19]. Additionally, tumor cells have altered mitochondrial gene expression, including increased ubiquinol-cytochrome C reductase complex assembly factor 3 expression, which contributes to increased hypoxia by promoting a positive feedback loop with mitochondrial reactive oxygen species (ROS) to stabilize HIF-1α expression and enhance glycolysis [20].

While aerobic glycolysis is primarily favored, tumor cells expand their energy production by involving protein biosynthesis and fatty acid synthesis as well. The mTOR pathway is known for regulating cell growth through facilitating biosynthesis of proteins, lipids and nucleic acids [21]. More specifically, fatty acid synthesis is critical to maintain membrane biosynthesis for rapid proliferation. Fatty acid synthesis in cancer is regulated by many factors, including: tumor suppressor p53 (TP53; p53), c-Myc, PI3K/Akt/mTOR pathway, AMPK, HIF-1α, sterol regulatory-element binding proteins (SREBPs) and ROS [22]. SREBP-1 regulates enzymes of the pentose phosphate pathway, the conversion of acetate and glutamine into acetyl-CoA and enzymes responsible for converting acetyl-CoA to fatty acids, specifically via activation of the mevalonate (MVA) pathway, to enhance glycolytic activity (Figure 2) [23].

## 2. The Mevalonate Pathway

### 2.1. Regulation of Metabolic Reprograming by Tumor Suppressor and Proto-Oncogenic Genes 

Tumorigenic processes, including cell proliferation, migration and angiogenesis, require significant energy, so cancer cells adopt fundamental changes in energy metabolism, nutrient uptake and substrate utilization. Most cancers are characterized by the acquisition of profound mutations in tumor suppressor genes (Rb, PTEN, BRCA1/BRCA2) [24,25,26] or proto-oncogenes (HER2/neu, Ras, Myc, EGFR) [27,28,29,30]. One of the main tumor suppressor genes involved in cancer is p53. P53 plays a critical role in maintaining genomic stability and is considered to be the “guardian of the genome” [31]. It is well established that more than half of human tumors exhibit mutations of the TP53 gene. Missense mutations in the TP53 gene often disrupt its tumor suppressive functions but also result in the acquisition of oncogenic gain-of-function cellular activities that promote cancer progression and metastasis [32].

Mutant p53 has been shown to interact with nuclear SREBP-2 and enhance transcription of genes of the MVA pathway (Figure 2) [33]. Expression and post-translational modifications of oncogenic small GTPases, such as Rho and Ras, are upregulated following p53-induced activation of the MVA pathway and drive many tumorigenic processes, such as cell proliferation, migration and metabolism [34,35]. Mutant p53 also directly activates small GTPases, such as RhoA and the downstream effector ROCK, to promote GLUT1 translocation, which enhances glucose uptake and glycolysis in the Warburg effect [36]. In addition to affecting glucose metabolism, mutant p53 enhances lipid synthesis and bioavailability, at least partly by binding to AMPK to inhibit its activity [37]. In cancer cells that develop gain-of-function TP53 mutations, the upregulation of the MVA pathway appears to confer an important survival advantage, and the cells become dependent on this pathway to maintain viability [38].

### 2.2. Mevalonate Pathway

The MVA pathway is a complex signaling pathway that is involved in the production of cholesterol, vitamin D, lipoproteins, dolichol, ubiquinone and isoprenoids [39], and the pathway is outlined in Figure 2. In cholesterol biosynthesis, two acetyl-CoA molecules in the cytosol condense to form acetoacetyl-CoA, which reacts with a third acetyl-CoA to generate 3-hydroxy-3-methylglutaryl-CoA (HMG-CoA), facilitated by HMG-CoA synthase 1 (HMGCS1). HMG-CoA can be reduced by HMG-CoA reductase (HMGCR), the rate-limiting enzyme of the pathway, and is regulated by feedback from sterols and non-sterol metabolites derived from MVA [40]. HMCCR activity catalyzes the phosphorylation of MVA acid (also known as MVA) by MVA kinase and through a series of steps is metabolized to isopentenyl-5-pyrophosphate (PP) [40]. Isopentenyl-PP and dimethylallyl-PP are then condensed to produce larger isoprenoid molecules, including geranyl-PP, farnesyl-PP and geranylgeranyl-PP.

Further, the condensation of two farnesyl-PP molecules produces squalene, resulting in the formation of sterols, which are then converted to cholesterol. Rapidly dividing cancer cells require elevated levels of cholesterol to support membrane biogenesis, lipid raft formation within the cell membrane and a number of cancer cell functions, including proliferation, migration and invasion [41,42].

In the non-cholesterol branch of the MVA pathway, farnesyl-PP and geranylgeranyl-PP are both essential for protein prenylation, which is required for the membrane localization, anchoring and activity of several signaling proteins [43]. Included in this list of proteins are small GTPases, which are small G proteins with a molecular weight of 20–30 kDa that act as molecular switches by switching between “GTP-activated” and “GDP-inactivated” forms [44].

The Hippo pathway is a potent tumor suppressor pathway that includes the oncogenes yes-associated protein 1 (YAP) and transcriptional coactivator with PDZ-binding motif (TAZ). When the Hippo pathway is activated, YAP and TAZ are phosphorylated, which inhibits their activity. YAP and TAZ are activated by the MVA pathway [45], and this interaction is also driven by mutant p53 [46]. Geranylgeranyl-PP promotes nuclear localization and transcriptional activity of YAP/TAZ to drive cellular metabolism [45,47]. The inhibition of the MVA pathway reduces YAP/TAZ nuclear localization and induction of transcription [45,48]. In opposition to the inhibitory influence of the Hippo pathway, the Rho family of GTPases activate YAP/TAZ [49] and facilitate YAP/TAZ activation induced by a number of G-protein-coupled receptors [50]. As Rho GTPases require isoprenylation for their function, they depend on the HMGCR enzyme and are an important link between the MVA pathway and cellular metabolism. A central function of Rho GTPases is to regulate the actin cytoskeleton within the cell. The actin cytoskeleton is implicated in numerous cellular processes, including cell division, endocytosis and chemotaxis [51], and is specifically involved in cancer cell migration and invasion [52]. MVA-induced dysregulation of Rho GTPase is an important contributor to cancer progression.

### 2.3. Mevalonate Pathway in the Tumor Microenvironment 

Epithelial mesenchymal transition (EMT) is the process whereby cells lose E-cadherin expression and differentiate into mesenchymal cells with stem-like features and enhanced migratory potential [53]. The inducers of EMT in the tumor microenvironment are complex, although the involvement of MVA signaling is suggested through its positive correlation with tumor metastasis in breast, gastric and prostate cancer, among others [40]. The mechanisms by which MVA exacerbates EMT and subsequent metastases in these tumors are multi-modal and showcase the complex involvement of MVA in a myriad of cell processes (Figure 3) [54]. Knockout studies suggest that YAP1, a transcriptional co-activator of the Hippo pathway, plays a substantive role in metastasis, although its function depends upon its location within the cell [55]. Members of the Rho family of GTPases have also been implicated in the metastatic process, and evidence shows that a MVA signaling results in overexpression of RhoA with resultant translocation of YAP1 from the cytoplasm to the nucleus [56,57]. The localization of YAP to the nucleus facilitates signaling and triggers increased proliferation, migration and invasion of osteosarcoma cells [57,58]. Recently, MVA signaling has been shown to support protein N-glycosylation at the endoplasmic reticulum and N-glycan remodeling at the Golgi—both of which are essential components of EMT and tumor metastasis (Figure 3) [59]. Likewise, Brindisi et al. demonstrated that the addition of MVA activates the estrogen-related receptor alpha (ERRα) pathway in four different breast cancer cell lines. This leads to enhanced expression of proteins downstream, such as proliferator-activated receptor gamma coactivator 1-alpha, HER2/neu, tumor protein D52 and NOTCH2, which are responsible for enhanced proliferation, motility and propagation of cancer stem-like cells (CSCs) (Figure 3) [60].

CSCs exhibit indefinite potential for self-renewal, which makes them key drivers of tumor initiation and metastases. These cells are thought to arise through subpopulations of stem cells, which are maintained by a mutated stem cell niche affording aberrant proliferation [61]. The presence of CSCs contributes to the epidemic of treatment failure following initial treatment success, where CSCs lie dormant and are more resistant to therapies, inevitably fueling relapse [62]. Despite the clear involvement of CSCs in cancer progression, there are still no clinically available drugs targeting these cells directly [63]. Members of the Ras superfamily, specifically Rho GTPases, are essential in maintaining stemness in hematopoietic, embryonic, neural, epidermal, mesenchymal and cancer stem cells and are upregulated following activation of the MVA pathway (Figure 3) [64]. Likewise, brain-tumor-initiating cells have enhanced MVA pathway activity [65]. Indeed, studies have shown that basal RhoA activity is required for stem cells to retain self-renewal capacity [66]. Further, pharmacological targeting of downstream component Rho-associated protein kinase has been shown to abolish the contractility and collagen degradation capacity of breast cancer and melanoma-derived CSCs [67], demonstrating that the Rho pathway is also a key player in CSC metastasis. This impact of MVA signaling on EMT and propagation of CSCs is further substantiated by pre-clinical use of MVA pathway blockers. While addition of MVA exacerbated osteosarcoma cell motility in vitro, a mouse model of osteosarcoma lung metastasis revealed that simvastatin reduced the metastatic potential [57]. In addition, a statin-based blockade in the MVA pathway was shown to reduce the self-renewal capacity of CSCs and hinder motility in models of breast cancer, as well as decrease the risk of recurrence in prospective cohort studies [68,69,70]. The link between MVA and metastatic potential may help explain the inverse relationships between statins and cancer-specific mortality rate in a myriad of tumor subtypes [54].

Another pivotal cell type in the process of metastasis are cancer-associated fibroblasts (CAFs). In normal development, fibroblasts are key producers of connective tissue and participate in wound healing by producing cytokines and chemokines, including TGF-β. In the context of the tumor microenvironment, cues such as DNA damage, physiological stressors and inflammatory signals arising from tumors, instruct the activation of CAFs. CAFs contribute to the production of matrix proteases, allowing for remodeling of the tumor matrix that creates paths for tumor cell invasion. CAF-induced matrix production also leads to advantageous consequences for tumors, including pro-survival and pro-proliferative signaling, collapsed blood vessels leading to hypoxia, reduced immune cell motility and metabolic dysregulation [71]. The adaptation of the tumor microenvironment to hypoxic conditions not only leads to metabolic reprograming of cancer cells but also stimulates CAFs to change their bioenergetics to glycolysis [72]. Understanding the metabolic changes in CAFs is thereby relevant in the context of their interaction with cancer cells. CAFs have been implicated in conferring multi-drug resistance in tumor cells [73]. In co-culture experiments utilizing spheroids, CAFs were shown to upregulate cholesterol and steroid biosynthesis in pancreatic cell lines, leading to anti-androgen treatment resistance [74]. Further, genome-wide expression profiling in normal human prostate stromal cells co-cultured with human prostate cancer cells demonstrated an overexpression of MVA pathway enzymes HMGCS1 and HMGCR (Figure 3). Immunohistochemical analysis of human tissue by way of microarray confirmed that HMGCS1 and HMGCR were overexpressed in prostate cancer stroma, suggesting that their expression may play a role in transition from organ-confined to metastatic disease [75].

While hyperactivity of the MVA pathway leads to malignant transformation, it is important to note that ablation of MVA signaling may impair the function and survival of cells that may have a role in cancer immune surveillance. Although resting T cells rely on oxidative phosphorylation, activated T cells shift to aerobic glycolysis, fueling MVA metabolism [76]. Since cholesterol is an essential component of membrane synthesis, this shift ensures the bioavailability of building blocks for T-cell proliferation, differentiation and effector function, with lymphocyte activation resulting in increased SREBP-2 activity (Figure 3) [77,78]. Similarly, the pharmacological inhibition of acetyl-CoA acetyltransferase 1, a cholesterol esterification enzyme, led to enhanced cholesterol in the membranes of CD8+ T cells, which resulted in enhanced T-cell receptor clustering and downstream signaling [79]. In fact, acetyl-CoA acetyltransferase 1 deficient CD8+ T cells were more efficient at controlling metastases compared to unmodified T cells in a mouse model of melanoma [79]. The non-sterol branch for protein prenylation has also been implicated in T-cell synapse formation, proliferation, migration and cytotoxic responses [76], emphasizing the importance of MVA signaling in T-cell effector function (Figure 3) [78]. MVA metabolism has also been shown to promote macrophage survival in a Rac1-dependent manner and enhance the immunostimulatory effect of granulocyte-macrophage colony-stimulating factor in macrophage priming [80,81,82]. The immunomodulatory effects of the MVA pathway have inspired research into the potential of statins as therapeutic agents in autoimmune disease [83].

### 2.4. The Mevalonate Pathway and Therapy Resistance 

Overexpression of SREBP-1 and SREBP-2 is a major contributor to therapy resistance in cancer (Figure 3). Lipogenesis, or the metabolic formation of fat, has also been linked to drug resistance in cancer patients. In EGFR mutant lung cancer patients, SREBP-1 expression promoted drug resistance mainly through upregulated and sustained lipogenesis [84]. Inhibiting SREBP-1 and its downstream constituents demonstrated re-sensitization of tumor cells to therapies [84]. Likewise, inhibiting BRAF signaling in therapy-responsive melanoma patients led to downregulated SREBP-1 expression and subsequent hinderance of lipogenesis [85]. Therapy-resistant cells had the capacity to restore this pathway, promote lipid saturation and protect melanoma tumor cells from ROS and lipid peroxidation mediated damage [85]. More importantly, inhibiting SREBP-1 through pharmacological intervention re-sensitized BRAF-mutant melanoma cells to BRAF inhibitor therapies, demonstrating that combinational SREBP-1 and BRAF inhibitors are more effective in targeting drug-resistant cells [85]. The process of tumor cells shedding into the bloodstream and forming circulating tumor cells initiates substantial oxidative stress, resulting in metastasis [86]. Circulating tumor cells in melanoma patients had substantial upregulated pathways involved in lipogenesis and iron homeostatic pathways, correlated to drug-resistant circulating tumor cells harboring BRAF mutations [86]. SREBP-2 is a regulator of lipogenesis and is capable of transcribing transferrin, a known iron carrier, as well as reducing iron clusters, ROS and lipid peroxidation, ultimately contributing to ferroptosis and therapy resistance [86].

In many cancers, the MVA pathway is activated to increase cell growth, survival and therapy resistance. MVA pathway activity was shown to be increased in lapatinib- and trastuzumab-resistant HER2+ breast cancer cells [87]. Treatment with lipophilic statins and zoledronic acid, an N-bisphosphonate, resulted in tumor cell apoptosis and inhibited the growth of resistant cells [87]. The ERRα pathway is known to facilitate metabolic switching [60]. Both MVA and cholesterol products can activate this pathway to increase cell proliferation and migration, generate cancer stem-like cells and produce lipid droplets [60]. These properties are associated with increased tumor growth, aggressiveness and metastasis, ultimately resulting in ERRα pathway-mediated drug resistance (Figure 3) [60].

The MVA pathway is also important for the prenylation of proteins. The MVA enzyme, geranylgeranyl transferase II, and its corresponding substrate, Rab11b, promote overexpression of Arf6 [88]. This ultimately fosters EMT of tumor cells, contributing to enhanced metastatic potential and drug resistance [88]. Together, the overexpression of Arf6, mesenchymal proteins and upregulated MVA signaling are all correlated with drug resistance and poor patient survival outcomes (Figure 3). Inhibiting MVA pathway signaling as well as geranylgeranyl transferase II inhibited tumor cell invasion and metastatic capability, and, therefore, contributed to reduced therapy resistance [88]. Cisplatin-resistant ovarian cancer cells have a cytoskeleton composed of long actin stress fibers and are structurally stiffer than cisplatin-sensitive cells [89]. This cell stiffness is mediated by Rho’s influence on actin remodeling, and the inhibition of Rho was shown to decrease cell stiffness and re-sensitized cells to cisplatin [89]. More specifically, elevated RhoB expression in EGFR-mutated lung cancer tumors was associated with poor response to EGFR-tyrosine kinase inhibitors [90]. The use of Akt inhibitor therapies induced tumor cell death and disease regression in RhoB-positive cells [90]. This demonstrated that RhoB/Akt signaling plays a role in cancer cell resistance to EGFR-tyrosine kinase inhibitors [90].

As previously mentioned, YAP from the Hippo pathway is activated in response to MVA pathway upregulation. Both YAP and TAZ overexpression are contributors to drug resistance in cancer (Figure 3). Overexpression of YAP was present in chemoresistant hepatocellular carcinoma cell lines, and when YAP expression was inhibited, these cells became more sensitive to chemotherapy mainly through autophagy-related cell death [91]. YAP inhibition also increased Rac-driven ROS and subsequently inactivated mTORC1 signaling [91]. YAP/TAZ and mTORC1 signaling, including the downstream target survivin, are constitutively inhibited following MVA pathway blockage [87]. However, the overexpression of YAP rescued survivin expression even when MVA signaling was inhibited [87]. This suggests that MVA pathway signaling provided alternative YAP/TAZ-mTORC1-survivin-mediated tumor cell survival and promoted therapy resistance [87]. YAP also directly contributes to an increased expression of COX-2 at a transcriptional level in colorectal cancer [92]. This resistance mechanism was aided by supportive effectors, including survivin [92]. Treatment with a YAP/COX-2 inhibitor significantly inhibited activation, induced apoptosis and decreased the viability of chemoresistant cancer cells [92].

Therapy resistance is heavily influenced by the MVA pathway and its downstream constituents. Targeting this pathway therapeutically may battle drug resistance and treat cancer patients more effectively.

## 3. Targeting the Mevalonate Pathway

Statins, bisphosphonates, geranylgeranyl transferase inhibitors, farnesyltransferase inhibitors and squalene synthase inhibitors are all classes of drugs that target the MVA pathway. Statins inhibit the pathway early on at the rate-limiting enzyme, HMG-CoA reductase, and are effective in lowering the circulating cholesterol levels in the blood and protecting against heart attacks and strokes [93]. Bisphosphonates are classified as non-nitrogen-containing or nitrogen-containing, with nitrogen-containing bisphosphonates (e.g., zoledronic acid, risedronate acid) being more effective and inhibiting farnesyl pyrophosphate synthase directly [94]. This subsequently prevents the post-translational modification of small GTPases, inhibiting protein prenylation, which facilitates the loss of osteoclast activity in bones [95]. Based on this mechanism, bisphosphonates are mainly prescribed in patients with Paget disease of the bone, osteoporosis, myelomas and bone metastases [94]. Farnesyltransferase inhibitors (e.g., lonafarnib, tipifarnib) were developed as anti-cancer agents to specifically disrupt Ras farnesylation [96]. Geranylgeranyl transferase inhibitors (e.g., GGTI-2148, GGTI-298, P61A6) were developed to target downstream oncogenesis of Ras and are potent in inducing apoptosis as well as G1 phase cell-cycle arrest [97]. Squalene synthase inhibitors (e.g., zaragozic acid, YM-53601) work similarly to statins to decrease the circulating low-density lipoprotein (LDL) cholesterol levels through the induction of hepatic LDL receptors [98]. Unlike statin drugs, squalene synthase inhibitors do not inhibit the upstream steps of the MVA pathway and the isoprenoid route.

### 3.1. Statin Drugs

Statin drugs are one of the most prescribed classes of drugs and are predominantly prescribed as a cholesterol-lowering medication to protect against the risk of heart attack and stroke [99]. Statins work by competitively binding to the catalytic domain of the rate-limiting enzyme, HMGCR, preventing HMG-CoA conversion to MVA (Figure 2) [100]. This halts the pathway, preventing further production of cholesterol and inhibiting the synthesis of isoprenoids. SREBPs are subsequently transported from the endoplasmic reticulum into the Golgi apparatus for cleavage, and the now active SREBPs enter the nucleus and bind to sterol regulatory elements and drive the transcription of the LDL receptor [101]. As a result of the rise in LDL receptor activity, more circulating LDL cholesterol is taken up by their receptors, which eventually lowers LDL levels in the blood [102]. Although all statins work in the same way, their structural variations influence how effectively they lower plasma cholesterol levels.

Lovastatin became the first FDA-approved statin in 1987 [103], and there are now currently seven approved statins: atorvastatin, fluvastatin, lovastatin, pitavastatin, pravastatin, rosuvastatin and simvastatin [104]. An eighth statin, cerivastatin, was taken off the market worldwide in 2001 and is no longer approved due to multiple reports of fatal rhabdomyolysis [105]. All statins can be classified as either type I fungal derived (lovastatin, pravastatin, simvastatin) or type II fully synthetic (atorvastatin, fluvastatin, pitavastatin, rosuvastatin). The two main functional differences between type I and II statins are their solubility and their ability to interact with and inhibit HMGCR [106]. Based on the structural differences between statins, type II synthetic statins are known to form greater interactions with HMGCR based on their more hydrophobic structure [107]. This is mainly because type I statins all contain a naphthalene side ring, whereas type II statins contain a fluorophenyl group and various hydrophobic side ring structures (Table 1). These hydrophobic side ring structures mimic the HMG-CoA substrate, strengthening the binding of type II statins to HMGCR [107]. Atorvastatin and rosuvastatin specifically have additional binding interaction sites with HMGCR, and rosuvastatin also has a polar interaction between its methane sulfonamide pyrimidine side chain and HMGCR [108]. As such, rosuvastatin is most efficacious in reducing HMGCR activity by almost 50% followed by atorvastatin, simvastatin and pravastatin [109].

With respect to their solubility, statins are sub-divided into lipophilic and hydrophilic groups (Table 1), with lipophilic statins (atorvastatin, simvastatin, lovastatin, fluvastatin and pitavastatin) using passive diffusion to enter cells and being widely distributed in different tissues throughout the body, whereas hydrophilic statins (rosuvastatin and pravastatin) being membrane impermeable and requiring active transport. Therefore, they have less tissue absorption and are mainly taken up by the liver specifically [106]. The circulating half-lives of statins range from 1 to almost 20 h. As the bulk of endogenous cholesterol synthesis occurs overnight, statins with short half-lives are typically taken in the evening to optimally lower cholesterol levels following their production (Table 1) [109].

Statins inhibit the MVA pathway and therefore endogenous cholesterol production, and they are currently approved as single-agent or combination therapies to reduce plasma cholesterol and reduce coronary heart disease [110]. Through their potent inhibition of the MVA pathway and our more recent understanding of the role of MVA signaling in promoting cancer, there is interest in repurposing statin drugs as a potential therapeutic strategy in treating various cancers.

### 3.2. Pre-Clinical In Vitro and In Vivo Studies

Many researchers have indulged in pre-clinical research to better understand the mechanism through which statins have shown effectiveness in a number of cancers.

Atorvastatin has been shown to inhibit cell adhesion, proliferation and invasion and induced apoptosis, G1 cell-cycle arrest and autophagy in ovarian and breast cancer cell lines [111,112]. Mechanistically, atorvastatin decreased VEGF and matrix metalloproteinase 9 and inhibited c-Myc expression to further increase cancer cell sensitivity to atorvastatin’s inhibitory effects [111]. Additionally, the use of atorvastatin in combination with other cancer therapies, including ionizing radiation, increased the percentage of apoptotic breast and lung cancer cells, inhibited proliferation and increased reactive oxygen production [113]. Similarly, a combination therapy with celecoxib inhibited the formation and growth of prostate tumors [114].

The use of fluvastatin in combination with many current anti-cancer therapies demonstrated promising results. Fluvastatin and cisplatin resulted in robust inhibition of proliferation, G2/M cell arrest, impaired expression of Ras GTPase proteins and increased early apoptosis in ovarian cancer cells [115]. Fluvastatin use in combination with the histone deacetylase inhibitor, vorinostat, induced apoptosis and inhibited cell growth through the activation of AMP-activated protein kinase in renal cancer cells [116]. Fluvastatin treatment in a murine model of pancreatic cancer also demonstrated a robust anti-tumor effect when used in combination with radiation therapy [117].

Lovastatin treatment in lung cancer cells has demonstrated increased apoptosis and inhibited G1 phase cell progression by increasing the cell-cycle checkpoint inhibitors p21 and p27 and decreasing cyclin D1 expression [118]. In an ovarian cancer mouse model, lovastatin inhibited the overexpression of RhoA and metastatic activity [119] Similarly, lovastatin therapy also decreased tumor formation and lung metastasis in a sarcomatoid mammary carcinoma mouse model [120].

In liver cancer cells, pitavastatin inhibited growth and colony formation, induced activated caspase 3 and 9 apoptosis and induced G1 phase cell-cycle arrest [121]. Additionally, liver cancer xenograft mice treated with pitavastatin had overall improved survival and decreased tumor growth [121]. Pitavastatin treatment in combination with gemcitabine synergistically inhibited pancreatic cancer cell proliferation and cell-cycle G1/S phases arrest through downregulated cyclin A2/CDK2 and upregulated p21 and p27 [122]. Furthermore, pancreatic cancer xenograft mice treated with gemcitabine and pitavastatin had inhibited tumor growth [122]. The use of pitavastatin with cisplatin in lung cancer xenograft mice inhibited tumor angiogenesis by suppression of tumor endothelial cell migration and morphogenesis [123]. In addition, pitavastatin suppressed Ras/Raf/MEK/ERK and PI3K/Akt/mTOR signaling and associated protein prenylation, both of which have been implicated in promoting cancer progression [123].

Pravastatin treatment on hepatocarcinoma significantly decreased cell proliferation compared to sorafenib only or a combination therapy both in vitro and in vivo [124]. Pravastatin also suppressed matrix metalloproteinase-2 and -9 activity, which are known for being critical to tumor invasiveness, and decreased lung metastasis [125]. Additionally, gastric and colorectal cancer patients are known to have significantly lowered ApoA1 levels, an anti-inflammatory high-density lipoprotein constituent. Pravastatin treatment elevated ApoA1 levels and subsequently decreased tumor cell proliferation [126]. Pravastatin alone or combined with doxorubicin also reduced tumor size, yielding more efficacious treatment outcomes [126].

Rosuvastatin treatment on prostate cancer cells inhibited cell proliferation and spheroid formation and inhibited EMT through increased E-cadherin expression and decreased vimentin and Zeb-1 expression [127]. Rosuvastatin also induced G1 phase arrest, decreased cell viability and increased caspase 3 levels in papillary thyroid cancer cells [128].

Simvastatin treatment in breast cancer cells increased the expression of miR-140-5p, a known tumor suppressor gene in many cancers, through activation of the transcription factor NRF1, resulting in overall anti-proliferative and pro-apoptotic effects [129]. Cervical cancer cells have demonstrated high levels of HMGCR and increased GTPase activity, rendering them more sensitive to simvastatin-induced inhibition of the MVA pathway compared to paclitaxel treatment alone [130]. Simvastatin in combination with other therapies decreased proliferation and increased apoptosis through depletion of geranylgeranyl pyrophosphate, inhibition of GTPase protein prenylation, suppression of Ras and RhoA downstream signaling and increased AMPK phosphorylation [130,131]. Simvastatin alone or in a combination was more efficacious and associated with upregulation and accumulation of HMGCR in cervical, breast and pancreatic cancer mouse models [130,132].

In vivo, varying statins (pravastatin, atorvastatin, simvastatin, lovastatin, rosuvastatin, fluvastatin and cerivastatin) were capable of inhibiting Ras protein translocation, with the exception of pravastatin, demonstrating anti-tumor effects of statins on pancreatic cancer xenograft mice [133]. Using multiple murine models of spontaneous breast cancer metastasis to the liver and lung organs, both atorvastatin and rosuvastatin demonstrated efficacy in reducing the proliferation of metastatic tumor cells but not primary breast tumor cells [134].

There is a large body of research that suggests all statins have some anti-cancer capabilities. As a result, additional research, particularly clinical trials, are necessary to further optimize the use of statins as a therapy in various cancers.

### 3.3. Statins and Cancer Incidence and Mortality

With statins so widely prescribed, many researchers have retrospectively investigated whether there is a link between statin use and cancer incidence. Many other diseases act as comorbidity risk factors in the development of cancer. For instance, patients with heart failure have a higher risk of developing cancer. Heart failure patients taking a statin had a 16% lower risk of cancer incidence [135]. This reduced cancer incidence coupled with reduced cancer-related mortality, however, was dependent on the duration that a patient took the statin drug [135]. Comorbidity factors, including type 2 diabetes and kidney failure, also contribute to an increased cancer risk. Both kidney failure patients on dialysis and diabetic patients taking a statin for a prolonged period of time also have reduced incidence of numerous cancers [136,137]. Statins also have preventative benefits in individuals with no comorbidity factors present. For instance, gastrointestinal cancers account for 25% of cancer incidences worldwide [138]. Statin users of at least 6 months had significantly lower risk of esophageal, stomach and colorectal cancer incidence as well as decreased cancer mortality [139]. Additionally, statin use has also been shown to be associated with decreased incidence in lung [140], breast [141], gynecologic [142], pancreatic [143] and prostate [144] cancers. Taken together, statins have a protective effect in preventing cancer incidence in individuals with or without comorbidity risk factors when taken for a prolonged period.

There is also compelling evidence to support the role of statins in reducing cancer mortality and improving patient survival. Lung cancer patients with tumors harboring a p53 mutation, which accounts for about 50% of all lung cancer cases, were responsive to statin therapy, with higher levels of cancer cell apoptosis, inhibited cell proliferation and more tightly controlled lipid raft regulation [145]. Similarly, in patients with lung tumors harboring EGFR mutations, the combinational use of statins with EGFR tyrosine kinase inhibitors increased patient overall survival [146].

Retrospective studies demonstrated that statin use contributed to a lower risk of mortality in breast cancer patients [147]. More specifically, long-term statin use of five years or greater in female breast cancer patients was significantly associated with improved overall survival as well as disease-free survival in all subtypes [148]. The use of statins post-cancer diagnosis demonstrated statins have little to no protective effects on mortality in breast cancer patients or in preventing a second cancer occurrence [149,150], demonstrating that the timing of statin administration is important to see positive benefits.

Systematic analysis of epidemiological studies demonstrated improved overall survival in pancreatic patients using statin therapies [151]. Additionally, lipophilic statin use post-diagnosis was accompanied with significantly longer overall survival in non-metastatic patients [152].

Nationwide retrospective analysis demonstrated that statin use post-diagnosis improved survival in women with epithelial ovarian cancer [153]. This is further supported by similar studies evaluating statin use in women recently diagnosed with ovarian cancer and an associated lower risk of mortality [154].

Overall, several studies suggest that statins may be effective when used as anti-cancer agents, which justifies additional research into their effectiveness in clinical trials for cancer patients.

### 3.4. Statins and Current Clinical Trials 

The repertoire of retrospective and prospective studies as well as pre-clinical in vitro and in vivo research presents a compelling tale to support the hypothesis that statins may be a useful adjunct to cancer therapy. As a result, a slew of clinical trials were launched to look into the matter further. Simvastatin combined with fluorouracil, adriamycin and cyclophosphamide therapy showed improvement in the objective response rate and pathological response in patients with locally advanced breast cancer [155]. The use of a statin (or metformin) contributed to improved survival time in pancreatic patients [151]. Quite unexpectedly, several other clinical trials found no improved overall survival nor additional benefits from adding a statin to the current systemic anti-cancer therapies [156,157,158].

Many of these trials were poorly designed due to various reasons, including: intervening with statins in late-stage cancer, evaluating statin efficacy in a variety of cancers, dosing used being based on treating hypercholesterolemia and not taking into account the half-life and dosing schedule of the statins used [159]. There may be many reasons contributing to this lack of efficacy, including: the dose used, the timing of intervention, the use of certain statins for specific cancers and a multitude of other factors. Currently, there are numerous clinical trials underway that are actively recruiting participants with more specifically targeted cancers and use of specific statins (Table 2). Finding the right combination of statins, dosage and malignancies that respond to them could improve patient outcomes.

## 4. Considerations When Targeting Cancer Metabolism with Statins

As tumor heterogeneity causes significant disparities in therapeutic responses within the tumor, often leading to recurrence or resistance [19], it is critical to understand these unique components of the tumor from a therapeutic perspective to optimize therapy efficacy. To take advantage of the addition of a statin to boost cancer therapeutic strategies, we must optimize and better understand the myriad of variables influencing statin drug efficacy. As a result, certain statins have been shown to be effective in specific subtypes of cancers. It is vital to keep in mind that some genetic abnormalities may have an impact on a statin’s reactivity.

Simvastatin specifically demonstrated efficacy in lung cancer cells harboring p53 missense mutations through inhibition of cancer cell growth, reduced lipid rafts and induced tumor cell apoptosis [145]. Similarly, rosuvastatin monotherapy had specificity to T-cell lymphomas harboring a p53 R248Q DNA contact mutation but not in tumor cells expressing an R172H p53 conformational mutation [160]. Additionally, metastatic colorectal cancer patients specifically harboring KRAS mutated recto-sigmoid cancers had an increased benefit from lipid lowering treatments, including statins, demonstrating not all colorectal patients will show the same efficacy to statin treatment [161]. Likewise, statin inhibition of KRAS specifically promotes immunogenic cell death specifically of KRAS mutant cancer cells [162]. This shows an extremely specialized cell death in KRAS mutant cancers.

In a similar line, the lipophilicity of the statin used may have an impact on cancer response. Lipophilic statins have a greater capability to passively diffuse through cellular membranes and can therefore penetrate both hepatocytes as well as non-hepatocytes [107,163]. Lipophilic statins also possess greater cytotoxic potential, especially through pro-apoptotic activity in cancer cells [164,165]. When compared to hydrophilic statins, this versatility makes lipophilic statins more adaptable, and it may be a significant feature to consider when deciding which statin to use.

The dose, timing its implementation and duration of usage for each statin may influence its efficacy. Many studies have significantly shown a strong link between the combinational use of statins and other cancer therapies proving to be efficacious, especially in reducing the risk of cardiotoxicity [166,167,168,169]. Alternatively, patients with a familial history of cancer risk may benefit from taking a statin as a preventative strategy in delaying or preventing the onset of cancer [170,171,172,173]. There is also the possibility of using statins as a maintenance monotherapy to prevent relapse in remission patients, albeit this has not been studied extensively. Statins, particularly lipophilic statins, are also known to have strong affinity for albumins, resulting in poor bioavailability and posing a challenge when using statins as a cancer therapy [174]. This has steered research into designing proper drug carriers in an attempt to overcome this major roadblock. As such, there are many potential applications of statins as a cancer therapeutic that still need to be explored. Statins as a cancer therapy will require a patient-by-patient approach to develop a specific, effective treatment plan.

Cancer is a whole-body disease, complicating other organ function with metastases leading to physiological imbalance. Cancer and heart disease are two leading causes of mortality, with numerous studies implicating tumor-induced changes as a major contributor to cardiac dysfunction in patients [175]. In fact, 60–80% of patients with advanced cancer exhibit cachexia, which often encompasses respiratory or heart failure, and 30% of cancer-related deaths are a result of this muscle-wasting syndrome [176]. Cardiac cachexia is thought to result from factors released by tumors, including inflammatory cytokines, lipolytic and proteolytic factors, which lead to progressive functional impairment of the cardiac muscle [177]. There is also a myriad of evidence implicating chemotherapy and other cancer therapeutics in exacerbating cachexia. Evidence also points to the involvement of metabolic derangements as a driver of cachexia [178]. In light of the influence of statins in targeting pro-inflammatory cytokines, improving endothelial cell function and inducing endothelial progenitor cells, researchers have suggested a use for statins in combatting cardiac cachexia [176], although clinical data for this indication of statin use are not yet available. Data supporting statin drugs improving chemotherapy or tumor-induced cardiac dysfunction would further merit statin use in the cancer setting.

In addition, given the benefit of statins in LDL cholesterol independent diseases, it has been proposed that the effects of these agents are pleiotropic, and we are just beginning to uncover the molecular pathways that they influence. In fact, in addition to the blockbuster drug’s influence on hypercholesterolemia, statins have had clinical significance in periodontal disease and rheumatoid arthritis with evidence of decreased inflammation [179,180], kidney disease with an impact on creatinine levels [181] and venous thromboembolism due to the influence on thrombosis and blood coagulation [182,183]. The clinical benefits of statins extend to pneumonia, multiple sclerosis, bone strength and gastrointestinal issues, often through unknown mechanisms of action [184].

There is compelling evidence that the use of statins as a potential cancer therapy should be investigated further in clinical trials. From a pharmaceutical perspective, de novo drug design is extremely time consuming, quite costly and is a high-risk process, with 90% of potential drug candidates failing in clinical trials [185,186]. On the other hand, drug repurposing is a low-risk strategy of taking already approved and existing market drugs and finding new therapeutic areas for their use [187]. One of the key advantages of repurposing a drug is that its safety profile has already been established because it has passed all Phase I, Phase II and Phase III clinical studies, which could lead to a priority phase III clinical evaluation [188].

Thalidomide has already been repurposed from its original approved use as a morning sickness medication to its present use as a multiple myeloma therapy [189]. Thalidomide monotherapy in refractory and relapsed patients showed partial response rates of 40% [190]. When combined with dexamethasone, thalidomide proved to be even more efficacious through patients demonstrating a high frequency of response to combination therapy, rapid remission and little to low incidence of irreversible drug toxicity [191,192]. Similarly, metformin, which was first approved to treat type II diabetes, has lately been repurposed as an anti-cancer drug [193]. Mechanistically, it is suspected that metformin activates the AMPK signaling and inhibits the mTOR pathway to elicit an anti-cancer response [194,195]. Metformin use as an anti-cancer agent has reduced cancer incidence, reduced cancer mortality, enhanced therapeutic efficacy when used in combination with radiotherapy and chemotherapy, reduced tumor malignancy, reduced cancer relapse and minimized cytotoxic and damaging effects [196]. As new research on statin drugs and support for their anti-cancer qualities continues to emerge, statins may be the next therapy to be repurposed for the treatment of cancer.

## 5. Conclusions

The combination of a heterogeneous tumor microenvironment and dysregulated metabolic pathways mediated by mutant tumor suppressor and proto-oncogenes promotes tumor survival and therapeutic resistance. The mevalonate pathway has been shown to be elevated in a variety of malignancies, impacting many cellular processes, and ultimately promoting tumorigenesis. As a result, the mevalonate pathway could be targeted by statins to inhibit tumor growth and metastasis. Despite the fact that several studies have been conducted over the last few decades to investigate the use of statins to treat various cancers, there has been limited concrete evidence of success in clinical trials. Statins may show efficacy in specific cancer types, after tailored dosage and timing of application, and when used in specific combination treatments. All of these issues should be taken into consideration in order to optimize the use of statin drugs as a cancer therapy and improve clinical success.

## Figures and Tables

**Figure 1 cancers-14-03500-f001:**
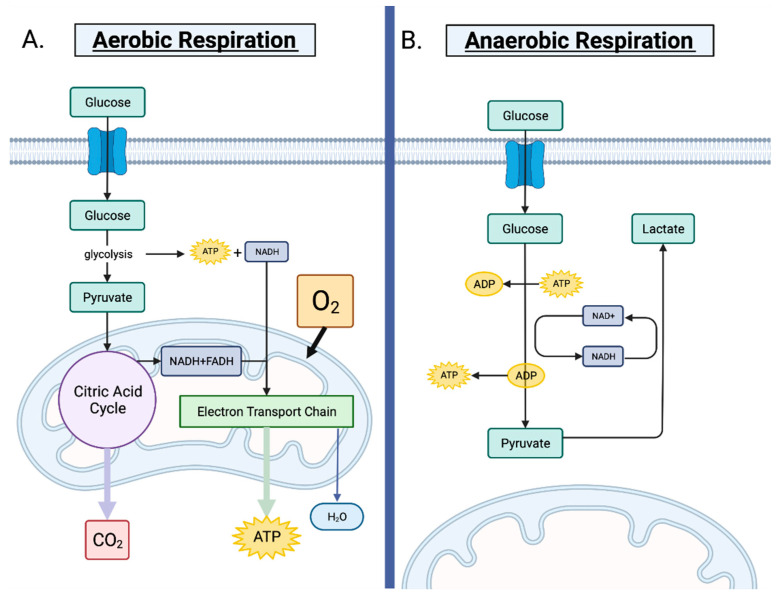
Aerobic respiration vs. anerobic respiration. (**A**). Normal, non-cancerous cells prefer aerobic respiration, which occurs in the presence of oxygen. The conversion of glucose to pyruvate is facilitated by glycolysis. The citric acid cycle then transforms pyruvate to acetyl-CoA in the mitochondria, also creating ATP, NADH and FADH. ATP is produced via oxidative phosphorylation of NADH and FADH, which fuels the electron transport chain. (**B**). When oxygen is scarce, cells switch to anaerobic respiration to generate ATP. Pyruvate cannot enter the citric acid cycle after glycolysis and is instead converted to lactate to avoid its accumulation. The NADH produced during glycolysis is converted back into NAD+, allowing glycolysis to continue. When oxygen levels return to baseline, cells resume aerobic respiration.

**Figure 2 cancers-14-03500-f002:**
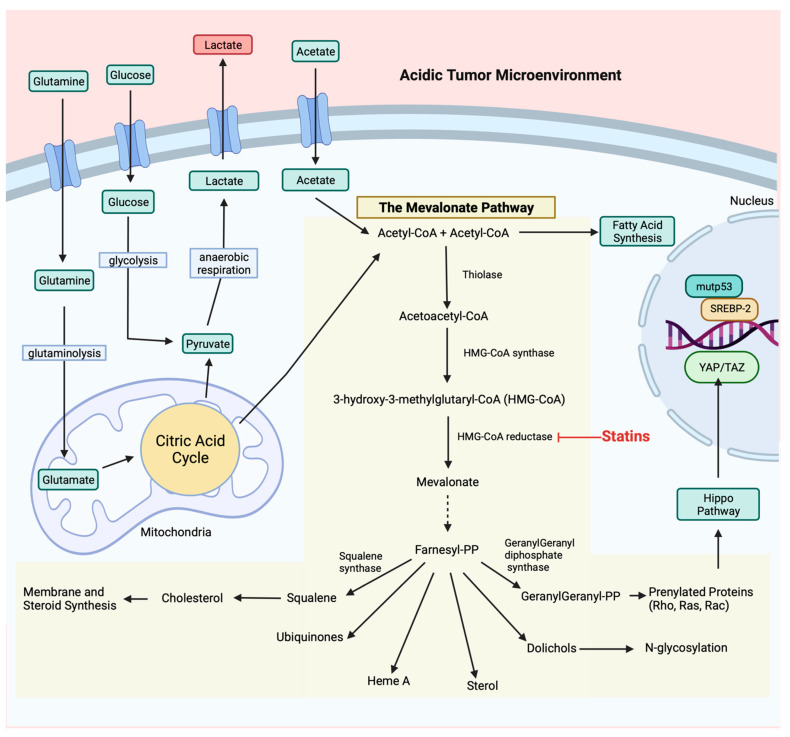
The regulation of the mevalonate (MVA) pathway and its role in promoting tumorigenesis. Tumor cells generate ATP through aerobic glycolysis and glutaminolysis, with lactate as a by-product. The extracellular accumulation of lactate contributes to an acidic and hypoxic tumor microenvironment. Mutant p53 interacts with SREBP-2 in the nucleus and dramatically increases MVA pathway enzyme transcription. The MVA route generates cholesterol, dolichols, ubiquinone and isoprenoids. Isoprenoids, particularly RhoA, inhibit the Hippo pathway, which prevents phosphorylation of YAP/TAZ. This enables YAP/TAZ to translocate to the nucleus and transcribe oncogenic target genes. Statin drugs competitively inhibit HMG-CoA reductase, the rate-limiting enzyme, of the MVA pathway.

**Figure 3 cancers-14-03500-f003:**
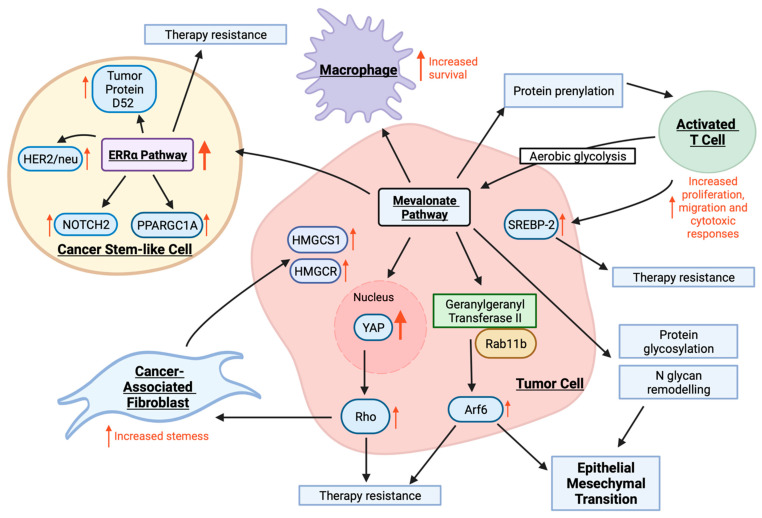
The pleiotropic effects of tumor cells, cancer stem-like cells (CSCs), cancer-associated fibroblasts (CAFs) and immune cells within the tumor microenvironment. The mevalonate (MVA) pathway upregulation in tumor cells contributes to increased protein prenylation and activation of GTPases, including Rho, and increased Hippo pathway activity, specifically YAP. MVA pathway activity, as well as Rho and YAP, contribute to epithelial to mesenchymal transition of tumor cells and therapy resistance. MVA pathway upregulation activates the ERRα pathway in CSCs, which in turn stimulates multiple proto-oncogenes, enhancing the propagation of CSCs and promoting therapy resistance. Rho activation increases stemness of CAFs, which also contribute to increased HMG-CoA synthase 1 (HMGCS1) and HMG-CoA reductase (HMGCR) expression, sustaining increased MVA pathway expression. Upregulated MVA signaling also alters immune cell function by stimulating T-cells’ shift to aerobic glycolysis, fueling MVA metabolism as well as increased SREBP-2 activity. The non-sterol branch of the MVA pathway is also implicated in increasing T-cell proliferation, migration and cytotoxic responses. Macrophage survival is also promoted in a Rac-1-dependent manner. Together, the various cell types within the tumor microenvironment contribute to the promotion of tumorigenesis.

**Table 1 cancers-14-03500-t001:** Molecular and solubility characteristics of the different statins.

Statin	Statin Type	Side Ring	Solubility	Transport	Half-Life (h)	Dose (mg/day)	Optimal Dosing Time
Atorvastatin	II	Pyrrole	Lipophilic	Passive	14	10–80	Any time of day
Fluvastatin	II	Indole	Lipophilic	Passive	1.2	20–80	Bedtime
Lovastatin	I	Naphthalene	Lipophilic	Passive	3	10–80	Morning and evening
Pitavastatin	II	Quinoline	Lipophilic	Passive	12	1–4	Any time of day
Pravastatin	I	Naphthalene	Hydrophilic	Active	1.8	10–80	Bedtime
Rosuvastatin	II	Pyrimidine	Hydrophilic	Active	19	5–40	Any time of day
Simvastatin	I	Naphthalene	Lipophilic	Passive	2	5–40	Evening

**Table 2 cancers-14-03500-t002:** Current active clinical trials investigating the effects of statin drugs on various cancers.

Statin	Cancer Type	Dose (mg/day)	Combination Agent	Trial Phase	Recruitment Status	Clinical Trial Number
Atorvastatin	Breast	40	Fulvestrant, Letrozole	Phase 2	Recruiting	NCT02958852
		80	NT	Phase 3	Recruiting	NCT04601116
		80	ACT/NCT	Phase 2; Phase 3	Recruiting	NCT05103644
		20–80	NCT	Phase 2	Recruiting	NCT03872388
	Colorectal	20	Aspirin	Phase 1	Recruiting	NCT04379999
		Not Specified	N/A	Phase 2	Recruiting	NCT04767984
	Head and Neck	20	ACT/NCT, ART/NRT	Phase 3	Not Recruiting	NCT04915183
	Hepatocellular	10	N/A	Phase 4	Recruiting	NCT03024684
	Pancreatic	80	Ezetimibe/Ezetrol, Evolocumab/Repatha, ACT	Phase 1	Recruiting	NCT04862260
	Prostate	80	ADT	Phase 3	Recruiting	NCT04026230
		80	Acetylsalicylic acid	Phase 3	Recruiting	NCT03819101
		Not Specified	Pentoxifylline, Vitamin E, ART/NRT	Phase 2	Recruiting	NCT03830164
	Solid tumor, AML	80	N/A	Phase 1	Recruiting	NCT03560882
Pitavastatin	AML, CLL	1–4	Venetoclax	Phase 1	Recruiting	NCT04512105
	Breast	2	NCT	Phase 2; Phase 3	Not Recruiting	NCT04705909
		40	ART	Phase 3	Recruiting	NCT04385433
		40	ART	Phase 2	Recruiting	NCT04356209
	AML	1280	ACT/NCT	Phase 2	Not Recruiting	NCT00840177
Rosuvastatin	Endometrial	10	Megestrol Acetate	Phase 2	Recruiting	NCT04491643
	Ovarian	Not Specified	Enoxaparin, Thromboprophylaxis	Phase 2	Not Recruiting	NCT03532139
	Melanoma, Solid tumor	10	Bupropion, ACT/NCT	Phase 1	Recruiting	NCT03864042
	Solid tumor	Not Specified	Sitravatinib, Nivolumab	Phase 1	Recruiting	NCT04887194
Simvastatin	Breast	40	ACT/NCT	Phase 2	Not Recruiting	NCT02096588
		20	N/A	Phase 3	Recruiting	NCT03971019
		80	HER2 Therapy	Phase 2	Recruiting	NCT03324425
		Not Specified	N/A	Phase 2	Not Recruiting	NCT03454529
	Liver	Not Specified	N/A	Phase 2	Not Recruiting	NCT02968810
	Lung	40	AZD9291	Phase 1	Not Recruiting	NCT02197234
	Ovarian	40	ACT/NCT	Phase 1	Recruiting	NCT04457089
	Lung	20	ACT/NCT	Phase 2	Recruiting	NCT04698941
		40	ACT/NCT	Phase 2	Not Recruiting	NCT04985201
		40	ACT/NCT	Phase 2	Not Recruiting	NCT01441349

Abbreviations: AML, acute myeloid leukemia; ACT, adjuvant chemotherapy; ART, adjuvant radiotherapy; ADT, androgen deprivation therapy; CLL, chronic lymphocytic leukemia; NCT, neoadjuvant chemotherapy; NRT, neoadjuvant radiotherapy; NT, neoadjuvant (therapy used not specified).

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
