# Peer review of "Mutant p53, the Mevalonate Pathway and the Tumor Microenvironment Regulate Tumor Response to Statin Therapy"

_cancers, 2022, doi:10.3390/cancers14143500_

Round 1
Reviewer 1 Report
The Review by Pereira et al., entitled “Characterizing the role of mutant p53, the mevalonate pathway and the tumor microenvironment to optimize statin therapy as an anti-cancer therapy” focuses on the relationship between mutant p53, mevalonate pathway and the use of statin in anti-cancer therapies.
This Review starts with a description of cancer cell metabolism and relative differences with normal cells, then offers an overview of the mevalonate pathway, its relationship with oncogenes, tumor microenvironment and chemoresistance. Then, the final paragraph on mevalonate pathway targeting covers statins and preclinical/clinical studies as anti-cancer drugs.
This is an overall hot topic, as competitive HMG-CoA reductase (HMGCR) inhibitors, statins not only control cholesterol levels, but also exhibit interesting lipid-independent pleiotropic effects. In particular, the controversial anti-cancer properties of statins have drawn oncologists attention, suggesting their potential use as repurposed drugs for cancer treatment.
The review is well structured, providing information about the relationship between oncogenes and the mevalonate pathway and the way to inhibit this pathway. However, it could be improved, as follows:
-The general language needs to be checked for accuracy and formality. For example, on line 353, the authors write: “Statins also competitively bind to the catalytic domain of the rate-limiting enzyme, HMG-CoA reductase, preventing its conversion to MVA (Figure 2)”. This implies that the enzyme itself will be converted in MVA and should be corrected in: “Statins also competitively bind to the catalytic domain of the rate-limiting enzyme, HMG-CoA reductase, preventing HMGCoA conversion to MVA (Figure 2)”.
-To avoid a simple list of papers and relative conclusions, it is desirable to integrate the knowledge provided with original observations, offering possible explanations and solutions to the interpretation of apparently conflicting studies. Regarding paragraph 3.2 on ”Statin and Cancer Incidence”, current literature offers a wealth of different studies about patients on statins, mostly bringing evidence of the statin protective role against different types of cancer. However, many studies, including the contraddictory ones, are simply listed and this confuses the reader. Furthermore, the authors mention 18 references (n. 109-126): of these, most are more than 10 years old and should be updated. More recent studies will help the reader to reach clearcut conclusions. A few examples among 2022 contributions on this subject: Cheung KS, et al. Statins associate with lower risk of biliary tract cancers: A systematic review and meta-analysis.Cancer Med. 2022 Jun 13.; Kim DS, et al Statins and the risk of gastric, colorectal, and esophageal cancer incidence and mortality: a cohort study based on data from the Korean national health insurance claims database.Cancer Res Clin Oncol. 2022 Jun 4.; Halámková J et al., Use of Hypolipidemic Drugs and the Risk of Second Primary Malignancy in Colorectal Cancer Patients.Cancers. 2022, 14(7):1699.
- As crucial topic to this review, statins should be described in more details. Although they are all inhibitors of Hydroxy Methyl Glutaryl-CoA Reductase (HMGCR), the main enzyme controlling mevalonate pathway that provides isoprenoids for prenylation of different proteins like the Ras superfamily, they might exert different roles depending on several factors. One such factor is the solubility profile of each compound: for example, the predominantly lipophilic statins (simvastatin, fluvastatin, lovastatin atorvastatin, pitavastatin,) can penetrate cells, whereas hydrophilic statins (pravastatin and rosuvastatin) rely on active transport and are more hepatoselective.
Therefore, cancer risk calculations, mortality rates or adverse effects etc in statin-treated patients could be affected by the nature of different statins used and may account for apparently contraddictory results in clinical studies. When available, information on the type of statin should be mentioned in the data description.
-I suggest to:
-expand the 3.1 “Statin drugs” paragraph with extensive mechanistic discussion on statins activities.
-include a table of the most commonly used statins, pinpointing their molecular and solubility differences (see also: Ward NC et al, Statin toxicity. Circulation Res, 124, n.2, 2019).
-“Preclinical in vivo and in vitro studies” (cellular) studies should follow 3.1 paragraph and become 3.2.
-Then, “Statin and cancer incidence” (former 3.2) and statin and cancer mortality (former 3.3) could be fused in a single paragraph, updating the references.
-Leave as last: “Statins and current clinical trials” (former 3.5).
Author Response
The Review by Pereira et al., entitled “Characterizing the role of mutant p53, the mevalonate pathway and the tumor microenvironment to optimize statin therapy as an anti-cancer therapy” focuses on the relationship between mutant p53, mevalonate pathway and the use of statin in anti-cancer therapies.
This Review starts with a description of cancer cell metabolism and relative differences with normal cells, then offers an overview of the mevalonate pathway, its relationship with oncogenes, tumor microenvironment and chemoresistance. Then, the final paragraph on mevalonate pathway targeting covers statins and preclinical/clinical studies as anti-cancer drugs.
This is an overall hot topic, as competitive HMG-CoA reductase (HMGCR) inhibitors, statins not only control cholesterol levels, but also exhibit interesting lipid-independent pleiotropic effects. In particular, the controversial anti-cancer properties of statins have drawn oncologists attention, suggesting their potential use as repurposed drugs for cancer treatment.
The review is well structured, providing information about the relationship between oncogenes and the mevalonate pathway and the way to inhibit this pathway. However, it could be improved, as follows:
-The general language needs to be checked for accuracy and formality. For example, on line 353, the authors write: “Statins also competitively bind to the catalytic domain of the rate-limiting enzyme, HMG-CoA reductase, preventing its conversion to MVA (Figure 2)”. This implies that the enzyme itself will be converted in MVA and should be corrected in: “Statins also competitively bind to the catalytic domain of the rate-limiting enzyme, HMG-CoA reductase, preventing HMGCoA conversion to MVA (Figure 2)”.
We have read through the entire review again and fixed any language/terminology discrepancies.
-To avoid a simple list of papers and relative conclusions, it is desirable to integrate the knowledge provided with original observations, offering possible explanations and solutions to the interpretation of apparently conflicting studies. Regarding paragraph 3.2 on ”Statin and Cancer Incidence”, current literature offers a wealth of different studies about patients on statins, mostly bringing evidence of the statin protective role against different types of cancer. However, many studies, including the contraddictory ones, are simply listed and this confuses the reader. Furthermore, the authors mention 18 references (n. 109-126): of these, most are more than 10 years old and should be updated. More recent studies will help the reader to reach clearcut conclusions. A few examples among 2022 contributions on this subject: Cheung KS, et al. Statins associate with lower risk of biliary tract cancers: A systematic review and meta-analysis.Cancer Med. 2022 Jun 13.; Kim DS, et al Statins and the risk of gastric, colorectal, and esophageal cancer incidence and mortality: a cohort study based on data from the Korean national health insurance claims database.Cancer Res Clin Oncol. 2022 Jun 4.; Halámková J et al., Use of Hypolipidemic Drugs and the Risk of Second Primary Malignancy in Colorectal Cancer Patients.Cancers. 2022, 14(7):1699.
We have condensed this section to make it more easier to read for the audience as well as updated the references to include more recent references.
- As crucial topic to this review, statins should be described in more details. Although they are all inhibitors of Hydroxy Methyl Glutaryl-CoA Reductase (HMGCR), the main enzyme controlling mevalonate pathway that provides isoprenoids for prenylation of different proteins like the Ras superfamily, they might exert different roles depending on several factors. One such factor is the solubility profile of each compound: for example, the predominantly lipophilic statins (simvastatin, fluvastatin, lovastatin atorvastatin, pitavastatin,) can penetrate cells, whereas hydrophilic statins (pravastatin and rosuvastatin) rely on active transport and are more hepatoselective.
Therefore, cancer risk calculations, mortality rates or adverse effects etc in statin-treated patients could be affected by the nature of different statins used and may account for apparently contraddictory results in clinical studies. When available, information on the type of statin should be mentioned in the data description.
We briefly mentioned these differences in 3.1 Statin drugs and have also expanded this section as well. We also discussed in detail in section 4 the different factors to take into account when using statins as an anti-cancer agent. If the specific statin used is not mentioned in the review that is because it was not indicated in cited papers.
-I suggest to:
-expand the 3.1 “Statin drugs” paragraph with extensive mechanistic discussion on statins activities.
-include a table of the most commonly used statins, pinpointing their molecular and solubility differences (see also: Ward NC et al, Statin toxicity. Circulation Res, 124, n.2, 2019).
-“Preclinical in vivo and in vitro studies” (cellular) studies should follow 3.1 paragraph and become 3.2.
-Then, “Statin and cancer incidence” (former 3.2) and statin and cancer mortality (former 3.3) could be fused in a single paragraph, updating the references.
-Leave as last: “Statins and current clinical trials” (former 3.5).
We have rearranged the various sub-sections in section 3 as indicated above, expanded section 3.1, included a table on all statins and their various properties, and condensed the statins and cancer incidence section and updated these references.
Reviewer 2 Report
Summary
The aim of this review paper is threefold: To characterise the role of mutant p53 and the mevalonate pathway in tumorigenesis, to provide an on overview of the role of the mevalonate pathway modifying the tumour microenvironment and contributing to treatment resistance, and to explore the current state of research into the use of stains as an adjuvant to current therapies. It clearly highlights the important role of the MVA pathway in the survival and metastatic potential of tumour cells and the potential for statins to interfere with these processes. It also outlines the current state of research at both the basic and translational level, and draws attention to the need for further research and more tailored approach to matching drugs to patients
General comments
Although there have been at least 4 major reviews in on the role of the MVA pathway in cancer and/or the potential therapeutic role for statins (as well as other inhibitors) published in the last two years, however the increasing number of studies and clinical trials suggest that this remains an area of significant interest and importance and another review is timely. It is clearly written and readable, covers the material well. Overall, I think that this review achieves the objectives you outline in your summary, and I only have a couple of points that I would suggest addressing.
1. Identifying that cancer cells with abnormalities in the MVA pathway could be sensitive to statins is certainly a generally accepted and plausible hypothesis. However, I do not think that your title accurately reflects the content of the paper. Firstly, the role of mutant p53 is mentioned only briefly, and p53 mutations are not the only oncogenic drivers that alter the MVA pathway (eg cMyc can also drive metabolic reprogramming and bind SREBP). The fact that gain-of-function p53 mutations functioning through the MVA pathway is very clear evidence of the importance of this pathway in tumorigenesis, I would not describe this section as ‘characterising’ the role of mutant p53, especially as these mutations have many other cellular effects in addition to those on cellular metabolism. Although you mention the use of rosuvastatin for T-cell lymphomas expressing p53R248Q but not R172H in section 4, you also discuss the specificity of statins for KRAS mutant cancers. You also do not mention the role of the MVA pathway in stabilising mutant p53 (another potential benefit of inhibiting the pathway).
Secondly, there is no mention anywhere how characterising the tumour microenvironment could optimise statin therapy.
These are not criticisms of the paper itself, but I would suggest altering the title and rewording the summary to reflect the review’s contents more accurately
2. In sections 3.2-3.3 you discuss what is known about statins and cancer incidence/mortality. Given the number of studies of varying sizes the results can often be contradictory, and it is very hard to provide a comprehensive overview. To this end I note that you have largely confined your discussion to large studies and meta-analyses, which is entirely appropriate, but several studies you cite are also more that 5-10 years old. Jeong et al. in 2019 and 2020 have published umbrella systematic reviews and meta-analyses on statins and cancer incidence https://doi.org/10.3390/jcm8060819 and mortality/survival https://doi.org/10.3390/jcm9020326 that support a number of the associations (or lack thereof) that you mention and potentially merit inclusion here.
3. In Figure 2 the details of the MVA pathway from mevalonate down to farenesyl-PP do not add additional information and are a distraction. I would suggest reducing the size of this figure by omitting some of this detail (and/or indicate the targets for the other drugs you mention at the start of part 3)
4. I would suggest an additional figure summarising the pleotropic effects of the activated pathways mentioned in the text sections 2.2-2.4, such as those that drive EMT, alter CAFs etc.
Minor comments
At line 457, do you mean inhibition of cMyc expression?
Author Response
Summary
The aim of this review paper is threefold: To characterise the role of mutant p53 and the mevalonate pathway in tumorigenesis, to provide an on overview of the role of the mevalonate pathway modifying the tumour microenvironment and contributing to treatment resistance, and to explore the current state of research into the use of stains as an adjuvant to current therapies. It clearly highlights the important role of the MVA pathway in the survival and metastatic potential of tumour cells and the potential for statins to interfere with these processes. It also outlines the current state of research at both the basic and translational level, and draws attention to the need for further research and more tailored approach to matching drugs to patients
General comments
Although there have been at least 4 major reviews in on the role of the MVA pathway in cancer and/or the potential therapeutic role for statins (as well as other inhibitors) published in the last two years, however the increasing number of studies and clinical trials suggest that this remains an area of significant interest and importance and another review is timely. It is clearly written and readable, covers the material well. Overall, I think that this review achieves the objectives you outline in your summary, and I only have a couple of points that I would suggest addressing.
- Identifying that cancer cells with abnormalities in the MVA pathway could be sensitive to statins is certainly a generally accepted and plausible hypothesis. However, I do not think that your title accurately reflects the content of the paper. Firstly, the role of mutant p53 is mentioned only briefly, and p53 mutations are not the only oncogenic drivers that alter the MVA pathway (eg cMyc can also drive metabolic reprogramming and bind SREBP). The fact that gain-of-function p53 mutations functioning through the MVA pathway is very clear evidence of the importance of this pathway in tumorigenesis, I would not describe this section as ‘characterising’ the role of mutant p53, especially as these mutations have many other cellular effects in addition to those on cellular metabolism. Although you mention the use of rosuvastatin for T-cell lymphomas expressing p53R248Q but not R172H in section 4, you also discuss the specificity of statins for KRAS mutant cancers. You also do not mention the role of the MVA pathway in stabilising mutant p53 (another potential benefit of inhibiting the pathway).
Secondly, there is no mention anywhere how characterising the tumour microenvironment could optimise statin therapy.
These are not criticisms of the paper itself, but I would suggest altering the title and rewording the summary to reflect the review’s contents more accurately
We have re-worded the title and changed some of the wording in the summary to better reflect the content of this review.
- In sections 3.2-3.3 you discuss what is known about statins and cancer incidence/mortality. Given the number of studies of varying sizes the results can often be contradictory, and it is very hard to provide a comprehensive overview. To this end I note that you have largely confined your discussion to large studies and meta-analyses, which is entirely appropriate, but several studies you cite are also more that 5-10 years old. Jeong et al. in 2019 and 2020 have published umbrella systematic reviews and meta-analyses on statins and cancer incidence https://doi.org/10.3390/jcm8060819and mortality/survival https://doi.org/10.3390/jcm9020326 that support a number of the associations (or lack thereof) that you mention and potentially merit inclusion here.
We have condensed this section and updated the references used.
- In Figure 2 the details of the MVA pathway from mevalonate down to farenesyl-PP do not add additional information and are a distraction. I would suggest reducing the size of this figure by omitting some of this detail (and/or indicate the targets for the other drugs you mention at the start of part 3)
We have updated this figure with a condensed outline of the mevalonate pathway.
- I would suggest an additional figure summarising the pleotropic effects of the activated pathways mentioned in the text sections 2.2-2.4, such as those that drive EMT, alter CAFs etc.
We have created a third figure to reflect the relationship between tumor cells, cancer-like stem cells, cancer-associated fibroblasts and immune cells within the tumor microenvironment.
Minor comments
At line 457, do you mean inhibition of cMyc expression?
We have corrected this wording.
Round 2
Reviewer 1 Report
This Reviewer is satisfied with the changes introduced by the authors. Two final checks before publication: 1. most references throughout the Review (not just in the Statin sections) are more than 6 years old. Please select the relevant ones and check whether newer citations could substitute the older ones. 2. please, check also language for informal/colloquial expressions, like "vs" abbreviation in paragraph title n. 1 (Non-cancerous vs Cancer Cell Metabolism) or "Things to consider.." in the title to paragraph n.4.
Author Response
This Reviewer is satisfied with the changes introduced by the authors. Two final checks before publication:
- most references throughout the Review (not just in the Statin sections) are more than 6 years old. Please select the relevant ones and check whether newer citations could substitute the older ones.
We have extensively updated the references throughout the review. There are sections of the review where we felt the use of an older citation was appropriate. For example, references 24-30 are mostly original articles when these proto-oncogene and tumor suppressor genes were discovered. We felt in these particular instances it was appropriate to use an original, older journal reference.
- please, check also language for informal/colloquial expressions, like "vs" abbreviation in paragraph title n. 1 (Non-cancerous vs Cancer Cell Metabolism) or "Things to consider.." in the title to paragraph n.4.
We have corrected this wording and terminology in these titles as well as made minor changes to some of the wording throughout the review as well.